# Blockchain Technology for Electronic Health Records

**DOI:** 10.3390/ijerph192315577

**Published:** 2022-11-24

**Authors:** Yujin Han, Yawei Zhang, Sten H. Vermund

**Affiliations:** 1Department of Biostatistics, Yale School of Public Health, Yale University, New Haven, CT 06510, USA; 2Department of Cancer Prevention and Control, National Cancer Center/National Clinical Research Center for Cancer/Cancer Hospital, Chinese Academy of Medical Sciences and Peking Union Medical College, Beijing 100021, China; 3Department of Epidemiology of Microbial Diseases, Yale School of Public Health, Yale University, New Haven, CT 06510, USA

**Keywords:** electronic health records, blockchain, interoperability, privacy

## Abstract

Compared with traditional paper-based medical records, electronic health records (EHRs) are widely used because of their efficiency, security, and reducing data redundancy. However, EHRs still manifest poor interoperability and privacy issues are unresolved. As a distributed ledger protocol composed of encrypted blocks of data organized in chains, blockchain represents a potential tool to solve the shortcomings of EHRs in terms of interoperability and privacy. In this paper, we define EHRs and blockchain technology and introduce several classic schemes based on blockchain technology to strengthen EHR interoperability and privacy protection. We then review ongoing challenges in the areas of data management efficiency, fairness of access, and trust in the systems. In this commentary, we suggest ongoing research needs for health informatics, data sciences, and ethics to establish EHRs based on blockchain technology. Blockchain-based EHR schemes must address the potential inequality of healthcare resources, the huge carbon footprint of computational needs, and potential distrust of health providers and patients that may ensue with wider use of blockchain technology.

An electronic health record (EHR) is the digital version of the patient’s health record which includes highly sensitive private information on history, diagnosis, and treatment. Other stored EHR data typically include appointments, billing and accounts, and laboratory tests. Indiana University, the Mayo Clinic, and Vanderbilt University were among the early academic EHR pioneers. In the 1970s, the federal government developed the country’s largest EHR system, VistA, for Veterans Affairs Health Care. By 1992, hardware had become cheaper and more powerful, and advancing computer technologies advanced the popularity of HER [1]. In 2004, EHR was incorporated into the Health Information Technology for Economic and Clinical Health Act (HITECH), reflecting the perceived need to convert medical records to EHR. New incentives and resources were available from the American Recovery and Reinvestment Act (ARRA or “ObamaCare”) beginning in 2009. The proportion of American hospitals using EHR systems had reached 95% by 2017 [2]. 

EHR reduces the inefficiency, insecurity, disorganization, data redundancy, and duplicate records problems associated with traditional paper-based medical records [3]. EHR is now regarded as an important part of the healthcare industry, though interoperability and privacy issues remain.

**Interoperability** may be defined as the degree to which computer software and/or systems share, interdigitate, and/or make use of data between programs and devices made by different manufacturers and/or engineers. For EHR, interoperability refers to the patients’ information sharing among different healthcare software systems. There are two factors leading to interoperability of EHR: (1) heterogeneities in hardware and software; (2) heterogeneities in the structure, purpose, and deployment of her software systems [4]. Efficient exchange of health information can be impeded with the use of differeherEHR systems using varying hardware and software systems, often with different data structures and purposes deployed in diverse healthcare institutions. Although interoperability standards are proposed by several organizations such as the European Committee for Standardization (ECS), International Organization for Standardization (IOS), Health Level Seven (HL7), Digital Imaging and Communications in Medicine (DICOM), and Integrating the Healthcare Enterprise (IHE), these standards have limitations. First, these standards are not established for wider healthcare and computing research communities; thus, understanding and applying these standards are difficult for non-specialists. Second, none of these standards cover aspects of EHR in a comprehensive fashion. For example, some standards are only on content and structure of EHR, while the others are only on EHR data access [4]. 

**Privacy** represents a second unresolved issue within EHR system deployment. Data privacy and security are increasingly important, given the ease with which digital data can be shared or stolen (i.e., “hacked”). Governments have collaborated to introduce laws and regulations to protect data security, such as the 73rd General Data Protection Regulation by the European Union and the California Consumer Privacy Act in the United States [5]. Nevertheless, data breaches of EHR are taking place at an unprecedented scale. An estimated 173 million medical records were breached between October 2009 and April 2017, affecting over half of the US population [6]. A study investigated the causes of EHR data breaches found that theft (47.5%) and loss (27.4%) were the major sources, indicating an urgent need for stricter control and limited access to patients’ sensitive information [7]. 

How can the interoperability and privacy issues of EHR be resolved? Blockchain technology is a potential tool.

**Blockchain** is database storage using encrypted blocks of data organized in chains for access as a distributed ledger protocol [8]. This means that all parties (nodes) participating in, contributing to, or accessing data in the blockchain decide together which blocks are validly accessed by different users through the use of consensus algorithms. Only a block of data that has been verified to be valid and relevant can be added into the data chain. In most systems, all nodes can have a copy of the complete blockchain so that they know when and what data were accessed and by whom.

Blockchain has advantages of decentralization, data transparency, and privacy and security [9]. Decentralization means that all data in blocks are distributed throughout the blockchain network, rather than stored in a central storage location. In addition, whether a block can be added to the chain is decided by all nodes together instead of by one central node. Figure 1 shows details of blockchain with a sequence of blocks. Due to the decentralization of data management, each node can have a complete copy of the blockchain such that all data access is completely transparent to every node in the blockchain, making it impossible to furtively tamper with data without knowledge of the other nodes. Privacy and security of blockchain means encrypting the data stored in the block with hash functions, such as the SHA-256 encryption algorithm. Cryptographic hashes are powerful one-way functions, and it is exceedingly difficult to reverse the plain text from the hash value, protecting blockchain from any third-party interference.

Blockchain is now well applied in the cryptocurrency economic sector. Its features of decentralization, transparency, and security also provide a potentially viable solution to the problem of poor interoperability and data leakage in EHR. A number of studies have proposed blockchain-based solutions, some of which we discuss here; we further seek to explain commonly used jargon in this field. We filter studies to be discussed in the paper based on the following strategy. It is important to note that this paper is not a systematic review, but rather a discussion of blockchain-based EHR solutions based on blockchain technology. Our aim in this paper is to discuss the popular blockchain-based EHR studies of the last few years, and these studies should focus on addressing the issues of interoperability or privacy. We included suitable publications based on the following search criteria: original research articles, conference proceedings; in English exclusively; and published between 2016 and 2021. We excluded duplicate articles; reviews, opinions, or surveys; articles without available full text; and articles without full text in English; and papers with too few citations (less than 500 citations).

To enable interoperability, the MedRec framework employs a complex series of “smart contracts” on an Ethereum blockchain between patients and visitors, including the registrar contract, the patient-provider relationship contract, and a summary contract to protect patient privacy and standardize the form of EHR [10]. In the MedRec, the exchange of data for the same patient in different medical jurisdictions is simplified by updating viewership permissions on the relevant data pointers of the Ethereum blockchain. Data pointers are cryptographic hash pointing to the data block that contain the data storage information and the cryptographic hash of the data. Such a uniform and simple operation enhances interoperability between different EHR systems. In addition, MedRec only stores data ownership and viewership permissions (cryptographic hash of the record) instead of raw medical records in blocks; thus, the security of user privacy data is also guaranteed.

The Ancile framework also deploys smart contracts in an Ethereum-based blockchain for heightened access control [11]. However, unlike MedRec, the Ancile blockchain sends the actual query link information in a private transaction over the Hypertext Transfer Protocol Secure (HTTPS). Using proxy re-encryption enables patients to remove access permissions and store keys and small encrypted records directly on the blockchain instead of storing the data locally. Ancile uses the QuorumChain open source consensus that achieves user agreement for what data are added and who can access which data by voting on new blocks by ”voter nodes”, in contrast to the “proof of work “ consensus that relies on network participants (“miners”) to calculate valid hashes.

OmniPHR is a distributed peer-to-peer (P2P) network to integrate EHRs, including a variety of technologies such as blockchain, routing overlay (a technology to decentralize data, locate nodes, and manage their location of P2P network), and Chord algorithm (a distributed lookup protocol, address how to efficiently locate the node of P2P network which stores a desired data item) [12]. OmniPHR allows patients to have a unified view of their health records and allows healthcare providers to access up-to-date data regarding their patients even if their data are stored at different healthcare institutions.

The Healthcare Blockchain System framework is designed for remote monitoring of patients and access to their health records. It uses a private blockchain based on the Ethereum protocol to protect patients’ privacy [13]. At the same time, it maintains a secure record of who has initiated activities on the blockchain and provides details about every data transaction. Notifications are delivered to all involved parties to address security vulnerabilities in remote patient monitoring. Combined with blockchain, MPC (Secure Multi-Party Computing), Indicator-Centric Schema (ICS), and Healthcare Data Gateway (HGD) frameworks all guarantee patients’ rights to own, control, and share their own data easily and securely [14]. MeDShare employs smart contracts and an access control to track the behaviors of patients’ privacy data and detect the possible invasion of patients’ privacy [15]. More specifically, MeDShare closely monitors all actions performed on the data through smart contracts and keys attached to the contracts, so that if a malicious user tries to steal data privacy or tamper with reports generated by the smart contract, their actions will be exposed and access will be restricted or even revoked. In addition to the above solutions, there are many more studies using blockchain technology to store, track, and manage medical records [16,17]. Researchers propose an IoMT-based platform for EHR based on the blockchain [17]. The method combines IoMT (Internet of Medical Things) and blockchain to encrypt and save the user’s health information. First, multiple smart sensors collect the user’s health recording, and then encrypted health data will be stored in the nodes of the Ethereum blockchain, thus protecting the privacy of users.

Blockchain shows promise to help EHR unify its standards, increase interoperability, and protect patient privacy. However, there are challenges to overcome with blockchain data access and sharing technology.

The first challenge is in **blockchain inefficiency**. Blockchain-based EHR necessitates complex computational operations, such as consensus mechanisms and digital signatures. These high-complexity operations impose an enormous computational burden on wireless nodes. Using Bitcoin cybercurrency as an example, the blockchain transaction processing mechanism is between 3.3 and 7 transactions per second, with the smallest transactions being 200–250 bytes [18]. Each transaction has to wait for about 10 min to be confirmed [19]. This is quite slow compared with the centralized systems used by banks that can deal with tens of thousands of transactions per second [19]. 

Similar to circumstances with cybercurrency data management, if a large number of patients and healthcare providers are added to the blockchain, the congestion caused by the huge computational burden can lead to query inefficiencies. Researchers simulated a private blockchain network consisting of one hospital node and 300 patient nodes [20]. The minimum mean propagation time (the average time it takes for a new block to reach most nodes in the network) was 1494.2 s and the minimum mean transaction time in the blockchain was 128.7 s. The results of these experiments pose the question whether an EHR system based on blockchain technology could meet the need for efficient and timely clinical, billing, or information sharing queries.

A second challenge concerns potential **injustice and inequity in access and high energy demands**. The blockchain uses incentive models to reward participation. In these models, transactions require network currency, such as Bitcoin or Ether. Currency can be earned by mining rewards or transaction fees. Network currencies have real value and they can be used for subsequent activities on the blockchain. In the MedRec, the rewards in the blockchain, the cryptocurrency Ether, will be given to care providers as a reward to incentivize them to participate in mining. The Ether blockchain protocol can be used for in-chain activities such as accepting viewing permissions and posting and updating contracts. Such an incentive is likely to cause medical inequities for lower resourced systems.

In a mechanism such as the proof-of-work (POW) blockchain system to obtain network currency, nodes (miners) in the blockchain need to constantly guess a random number (nonce) to obtain the power to generate the next block. If the hash value corresponding to the string obtained by combining the random number and the existing data of the blockchain meets the difficulty value requirements of the blockchain, the node can broadcast its calculation results, obtain the proof of work and the reward (network currency) of the corresponding block. Therefore, POW requires the care provider to perform repetitive calculations using specialized computer hardware until one gets the correct nonce. This energy-intense calculation has very high requirements for computing hardware and energy power [21], even requiring specialized computer hardware known as application-specific integrated circuits (ASICs) for the mining [22]. 

The **high energy requirements** for mining require massive additional computing capacity. Take the example of the Ether to be mined in MedRec; it takes an average of 7 MJ to generate just one US$, which is higher than the energy consumption of conventional mining copper and gold for the same value [23]. Healthcare providers with better computer hardware and more energy resources are likely to have more ownership in blockchain-based EHR systems. Given the dominance of the use of fossil fuels to generate electricity, the very high computational energy burden of blockchain will contribute to global greenhouse gas emissions and climate change. This system will reflect, and even amplify, the real-world inequalities in access to and use of valuable healthcare resources.

MedRec has proposed an alternative option, using the patient’s private data as a reward instead of Ether. This mode of reward risks injustice to patients. Without the protection from technologies such as differential privacy which is a cryptographic mechanism that can ensure almost consistent query results on adjacent data sets, the private information of patients remains at risk of leakage, even if the information is anonymous, especially for patients with rare diseases. Such an incentive model would be accompanied by further controversies about whether patients have the right to be informed about their own anonymized data, and whether this anonymization reward requires the patient’s consent.

One survey suggests that rare disease patients are willing to share their data with specific privacy requirements [23]. Blockchains such as Ancile blockchain bypass the risk of “currency” of patients’ privacy based on the QuorumChain consensus. However, this solution, which is based on a majority voting method to achieve consensus, is not secure enough at present. Malicious block-maker nodes can easily create inconsistencies and this protocol cannot ensure consensus in any realistic sense [24]. In addition, the QuorumChain consensus mechanism has not resolved how to fairly assign the roles of voter, maker, and observer to individual nodes. Hence, blockchain technology-based EMR sharing is not now equally accessible and affordable to all. Without improvements, the blockchain system has the potential to further amplify the uneven distribution of real-world healthcare resources.

A final challenge concerns trust issues for patients and healthcare workers alike. Even though smart contracts allow healthcare providers to trace requests for data, the lack of a regulatory mechanism may continue to raise concerns about inappropriate access and use of patient privacy information among healthcare providers and patients. The decentralized and transparent nature of blockchain makes data sharing safer and more secure, but also makes data regulation urgent. The lack of real-world regulation and lack of understanding of blockchain technology may lead to a crisis of trust between doctors and patients, and between different healthcare providers.

In conclusion, blockchain technology brings a new potential solution to the interoperability and privacy problems faced by EHR. It enables patients with one-stop shopping access to their medical histories and takes the burden of lifetime medical data stewardship from healthcare providers. However, existing blockchain-based EHR must address issues of efficiency, fairness, and trust. The integration of blockchain technology and EHR is not merely a technical issue that needs the attention of data scientists. For blockchain to reach its full potential towards EHR and other public health applications, data science, public health, medical, and nursing communities must address, through research and development, the inefficiencies and healthcare inequities potentially being exacerbated by blockchain-based EHRs, the carbon footprint on the environment due to high energy consumption created by blockchain-based EHRs, and the distrust of blockchain technology by healthcare providers and patients.

## Figures and Tables

**Figure 1 ijerph-19-15577-f001:**
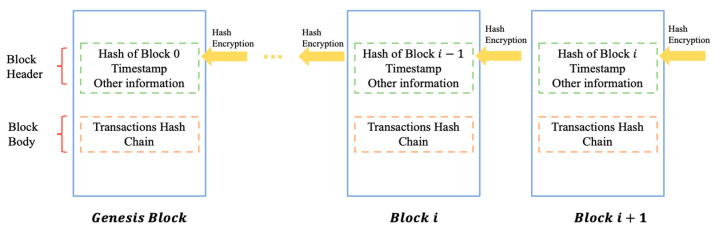
An example of blockchain which consists of a continuous sequence of blocks.

## Data Availability

Not applicable.

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
