# Peer review of "Blockchain Technology for Electronic Health Records"

_ijerph, 2022, doi:10.3390/ijerph192315577_

Round 1
Reviewer 1 Report
Authors provide a review of the use of blockchain technology in the frame of EHRs focusing on the usefulness for EHR interoperability and privacy protection. They suggest ongoing research needs for health informatics, data sciences, and ethics in order to establish EHRs based on blockchain technology.
The topic of the application of blockchain technology to the healthcare domain, and in particular to improve EHR Interoperability and data and privacy protection is of interest and widely treated in the last few years. Author provide a clear review of the major applications and freameworks using this technology and analyse features, potential solutions to EHR interoperability and address major issues connected to the topic.
Eventhough, it is not clear what is the metodology followed to exectute the review.
The paper does not include any kind of structure (even if the instructions for authors give explicite indications, i.e.), so it is not clear where finish the introduction, where start or finish the method and results, etc.
The language is clear and content understandeable also to a non expert reader, but seems that the paper is not really complete and well structured.
When you talk about interoperability standards, and cite, HL7, DICOM, etc., you can cite also IHE.
Is the Etherum protocol different from the Ether one cited at page 4 (line 158)? It is not clear when you cite them.
Also references lack of some informations (authors name, year of publication, publisher, etc.), see for example references n.3, 4, 17, etc.
How your review differs from the others available on that topic?
Typos:
Line 39: computer software and/and systems -> and/or systems
Reviewer 2 Report
The position paper suggests using blockchain and several established blockchain-based schemes to potentially address the issues with electronic health records (EHRs). The article also discusses issues with data management effectiveness, access equity, and system trust. Strengthening EHR interoperability and privacy protection is the ultimate objective.
With the help of definitions for terms like interoperability, privacy, and blockchain, the paper gradually introduces the reader to the vocabulary. The paper ends with a brief conclusion after discussing a few difficulties with blockchain usage. The information presented in the article is well-organized and concise, and the writing style could be described as academic. I think this article has the potential to influence and inspire further blockchain development in the field of electronic health records. The paper has enough recent references for its length, demonstrating that the subject was thoroughly researched and documented.
The authors should add more related work regarding IoT integration in EHR and fraud on EHR, for example:
- Bratulescu, Razvan-Alexandru, et al. "Fraudulent Activities in the Cyber Realm: DEFRAUDify Project: Fraudulent Activities in the Cyber Realm: DEFRAUDify Project." Proceedings of the 17th International Conference on Availability, Reliability and Security. 2022.
- Jain, Megha, Dhiraj Pandey, and Krishna Kewal Sharma. "A Granular Access-Based Blockchain System to Prevent Fraudulent Activities in Medical Health Records." Advances in Data Computing, Communication and Security. Springer, Singapore, 2022. 635-645.
- Ktari, Jalel, et al. "IoMT-based platform for E-health monitoring based on the blockchain." Electronics 11.15 (2022): 2314.
I recommend adding some diagrams or graphs to support the idea presented and make it easy for the reader to understand the concepts discussed.
Reviewer 3 Report
Electronic technologies certainly bring many advantages in terms of efficiency, security and reduction of data redundancy, especially the use of electronic health records (EHR). However, EHRs still manifest poor interoperability and there remain security and privacy issues to be resolved. Certainly blockchain technology may be a candidate for solving many of these problems. This article proposes blockchain-based EHR schemes should address the potential inequity of healthcare resources, the huge carbon footprint of computational needs, and the potential distrust of healthcare providers and patients that may arise with broader use of blockchain technology.
Author Response
Thanks for your comments and reading!